# Adaptive Bayesian Sampling with Monte Carlo EM

**Anirban Roychowdhury,   Srinivasan Parthasarathy**
Department of Computer Science and Engineering
The Ohio State University
roychowdhury.7@osu.edu, srini@cse.ohio-state.edu

## Abstract

We present a novel technique for learning the mass matrices in samplers obtained from discretized dynamics that preserve some energy function. Existing adaptive samplers use Riemannian preconditioning techniques, where the mass matrices are functions of the parameters being sampled. This leads to significant complexities in the energy reformulations and resultant dynamics, often leading to implicit systems of equations and requiring inversion of high-dimensional matrices in the leapfrog steps. Our approach provides a simpler alternative, by using existing dynamics in the sampling step of a Monte Carlo EM framework, and learning the mass matrices in the M step with a novel online technique. We also propose a way to adaptively set the number of samples gathered in the E step, using sampling error estimates from the leapfrog dynamics. Along with a novel stochastic sampler based on Nosé-Poincaré dynamics, we use this framework with standard Hamiltonian Monte Carlo (HMC) as well as newer stochastic algorithms such as SGHMC and SGNHT, and show strong performance on synthetic and real high-dimensional sampling scenarios; we achieve sampling accuracies comparable to Riemannian samplers while being significantly faster.

## 1   Introduction

Markov Chain Monte Carlo sampling is a well-known set of techniques for learning complex Bayesian probabilistic models that arise in machine learning. Typically used in cases where computing the posterior distributions of parameters in closed form is not feasible, MCMC techniques that converge reliably to the target distributions offer a provably correct way (in an asymptotic sense) to draw samples of target parameters from arbitrarily complex probability distributions. A recently proposed method in this domain is Hamiltonian Monte Carlo (HMC) [1, 2], that formulates the target density as an "energy function" augmented with auxiliary "momentum" parameters, and uses discretized Hamiltonian dynamics to sample the parameters while preserving the energy function. The resulting samplers perform noticeably better than random walk-based methods in terms of sampling efficiency and accuracy [1, 3]. For use in stochastic settings, where one uses random minibatches of the data to calculate the gradients of likelihoods for better scalability, researchers have used Fokker-Planck correction steps to preserve the energy in the face of stochastic noise [4], as well as used auxiliary "thermostat" variables to control the effect of this noise on the momentum terms [5, 6]. As with the batch setting, these methods have exploited energy-preserving dynamics to sample more efficiently than random walk-based stochastic samplers [4, 7, 8].

A primary (hyper-)parameter of interest in these augmented energy function-based samplers in the "mass" matrix of the kinetic energy term; as noted by various researchers [1, 3, 6, 8, 9], this matrix plays an important role in the trajectories taken by the samplers in the parameter space of interest, thereby affecting the overall efficiency. While prior efforts have set this to the identity matrix or some other pre-calculated value [4, 5, 7], recent work has shown that there are significant gains to be had in efficiency as well as convergent accuracy by reformulating the mass in terms of the target parameters to be sampled [3, 6, 8], thereby making the sampler sensitive to the underlying geometry. This is done by imposing a positive definite constraint on the adaptive mass, and using it as the metric of

the Riemannian manifold of probability distributions parametrized by the target parameters. This constraint also satisfies the condition that the momenta be sampled from a Gaussian with the mass as the covariance. Often called Riemannian preconditioning, this idea has been applied in both batch [3] as well as stochastic settings [6, 8] to derive HMC-based samplers that adaptively learn the critically important mass matrix from the data.

Although robust, these reformulations often lead to significant complexities in the resultant dynamics; one can end up solving an implicit system of equations in each half-step of the leapfrog dynamics [3, 6], along with inverting large $\left(O(D^2)\right)$ matrices. This is sometimes sidestepped by performing fixed point updates at the cost of additional error, or restricting oneself to simpler formulations that honor the symmetric positive definite constraint, such as a diagonal matrix [8]. While this latter choice ameliorates a lot of the added complexity, it is clearly suboptimal in the context of adapting to the underlying geometry of the parameter space. Thus we would ideally need a mechanism to robustly learn this critical mass hyperparameter from the data without significantly adding to the computational burden.

We address this issue in this work with the Monte Carlo EM (MCEM) [10, 11, 12, 13] framework. An alternative to the venerable EM technique, MCEM is used to locally optimize maximum likelihood problems where the posterior probabilities required in the E step of EM cannot be computed in closed form. In this work, we perform existing dynamics derived from energy functions in the Monte Carlo E step while holding the mass fixed, and use the stored samples of the momentum term to learn the mass in the M step. We address the important issue of selecting appropriate E-step sampling iterations, using error estimates to gradually increase the sample sizes as the Markov chain progresses towards convergence. Combined with an online method to update the mass using sample covariance estimates in the M step, this gives a clean and scalable adaptive sampling algorithm that performs favorably compared to the Riemannian samplers. In both our synthetic experiments and a high dimensional topic modeling problem with a complex Bayesian nonparametric construction [14], our samplers match or beat the Riemannian variants in sampling efficiency and accuracy, while being close to an order of magnitude faster.

## 2 Preliminaries

### 2.1 MCMC with Energy-Preserving Dynamics

In Hamiltonian Monte Carlo, the energy function is written as

$$H(\boldsymbol{\theta}, \mathbf{p}) = -\mathcal{L}(\boldsymbol{\theta}) + \frac{1}{2}\mathbf{p}^T M^{-1} \mathbf{p}. \tag{1}$$

Here $\mathbf{X}$ is the observed data, and $\boldsymbol{\theta}$ denotes the model parameters. $\mathcal{L}(\boldsymbol{\theta}) = \log p(\mathbf{X}|\boldsymbol{\theta}) + \log p(\boldsymbol{\theta})$ denotes the log likelihood of the data given the parameters along with the Bayesian prior, and $\mathbf{p}$ denotes the auxiliary "momentum" mentioned above. Note that the second term in the energy function, the kinetic energy, is simply the kernel of a Gaussian with the mass matrix $M$ acting as covariance. Hamilton's equations of motions are then applied to this energy function to derive the following differential equations, with the dot accent denoting a time derivative:

$$\dot{\boldsymbol{\theta}} = M^{-1}\mathbf{p}, \quad \dot{\mathbf{p}} = \nabla\mathcal{L}(\boldsymbol{\theta}).$$

These are discretized using the generalized leapfrog algorithm [1, 15] to create a sampler that is both symplectic and time-reversible, upto a discretization error that is quadratic in the stepsize.

Machine learning applications typically see the use of very large datasets for which computing the gradients of the likelihoods in every leapfrog step followed by a Metropolis-Hastings correction ratio is prohibitively expensive. To address this, one uses random "minibatches" of the dataset in each iteration [16], allowing some stochastic noise for improved scalability, and removes the Metropolis-Hastings (M-H) correction steps [4, 7]. To preserve the system energy in this context one has to additionally apply Fokker-Planck corrections to the dynamics [17]. The stochastic sampler in [4] uses these techniques to preserve the canonical Gibbs energy above (1). Researchers have also used the notion of "thermostats" from the molecular dynamics literature [9, 18, 19, 20] to further control the behavior of the momentum terms in the face of stochastic noise; the resulting algorithm [5] preserves an energy of its own [21] as well.

### 2.2 Adaptive MCMC using Riemannian Manifolds

As mentioned above, learning the mass matrices in these MCMC systems is an important challenge. Researchers have traditionally used Riemannian manifold refomulations to address this, and integrate

the updating of the mass into the sampling steps. In [3] the authors use this approach to derive adaptive variants of first-order Langevin dynamics as well as HMC. For the latter the reformulated energy function can be written as:

$$H_{gc}(\boldsymbol{\theta}, \mathbf{p}) = -\mathcal{L}(\boldsymbol{\theta}) + \frac{1}{2}\mathbf{p}^T \mathbf{G}(\boldsymbol{\theta})^{-1}\mathbf{p} + \frac{1}{2}\log\left\{(2\pi)^D |\mathbf{G}(\boldsymbol{\theta})|\right\},\qquad(2)$$

where $D$ is the dimensionality of the parameter space. Note that the momentum variable $\mathbf{p}$ can be integrated out to recover the desired marginal density of $\boldsymbol{\theta}$, in spite of the covariance being a function of $\boldsymbol{\theta}$. In the machine learning literature, the authors of [8] used a diagonal $\mathbf{G}(\boldsymbol{\theta})$ to produce an adaptive variant of the algorithm in [7], whereas the authors in [6] derived deterministic and stochastic algorithms from a Riemannian variant of the Nosé-Poincaré energy [9], with the resulting adaptive samplers preserving symplecticness as well as canonical system temperature.

### 2.3 Monte Carlo EM

The EM algorithm [22] is widely used to learn maximum likelihood parameter estimates for complex probabilistic models. In cases where the expectations of the likelihoods required in the E step are not tractable, one can use Monte Carlo simulations of the posterior instead. The resulting Monte Carlo EM (MCEM) framework [10] has been widely studied in the statistics literature, with various techniques developed to efficiently draw samples and estimate Monte Carlo errors in the E step [11, 12, 13]. For instance, the expected log-likelihood is usually replaced with the following Monte Carlo approximation: $Q(\boldsymbol{\theta}|\boldsymbol{\theta}^t) = \frac{1}{m}\sum_{l=1}^{m}\log p(\mathbf{X}, \mathbf{u}_l^t|\boldsymbol{\theta})$, where $\mathbf{u}$ represents the latent augmentation variables used in EM, and $m$ is the number of samples taken in the E step. While applying this framework, one typically has to carefully tune the number of samples gathered in the E step, since the potential distance from the stationary distribution in the early phases would necessitate drawing relatively fewer samples, and progressively more as the sampler nears convergence.

In this work we leverage this MCEM framework to learn $M$ in (1) and similar energies using samples of $\mathbf{p}$; the discretized dynamics constitute the E step of the MCEM framework, with suitable updates to $M$ performed in the corresponding M step. We also use a novel mechanism to dynamically adjust the sample count by using sampling errors estimated from the gathered samples, as described next.

## 3 Mass-Adaptive Sampling with Monte Carlo EM

### 3.1 The Basic Framework

Riemannian samplers start off by reformulating the energy function, making the mass a function of $\boldsymbol{\theta}$ and adding suitable terms to ensure constancy of the marginal distributions. Our approach is fundamentally different: we cast the task of learning the mass as a maximum likelihood problem over the space of symmetric positive definite matrices. For instance, we can construct the following problem for standard HMC:

$$\max_{M\succ 0}\quad \mathcal{L}(\boldsymbol{\theta}) - \frac{1}{2}\mathbf{p}^T M^{-1}\mathbf{p} - \frac{1}{2}\log|M|.\qquad(3)$$

Recall that the joint likelihood is $p(\boldsymbol{\theta}, \mathbf{p}) \propto \exp(-H(\boldsymbol{\theta}, \mathbf{p}))$, $H(\cdot, \cdot)$ being the energy from (1). Then, we use correct samplers that preserve the desired densities in the E step of a Monte Carlo EM (MCEM) framework, and use the obtained samples of $\mathbf{p}$ in the corresponding M step to perform suitable updates for the mass $M$. Specifically, to wrap the standard HMC sampler in our framework, we perform the generalized leapfrog steps [1, 15] to obtain proposal updates for $\boldsymbol{\theta}, \mathbf{p}$ followed by Metropolis-Hastings corrections in the E step, and use the obtained $\mathbf{p}$ values in the M step. The resultant adaptive sampling method is shown in Alg. 1.

Note that this framework can also be applied to stochastic samplers that preserve the energy, upto standard discretization errors. We can wrap the SGHMC sampler [4] in our framework as well, since it uses Fokker-Planck corrections to approximately preserve the energy (1) in the presence of stochastic noise. We call the resulting method SGHMC-EM, and specify it in Alg. 3 in the supplementary.

As another example, the SGNHT sampler [5] is known to preserve a modified Gibbs energy [21]; therefore we can propose the following max-likelihood problem for learning the mass:

$$\max_{M\succ 0}\quad \mathcal{L}(\boldsymbol{\theta}) - \frac{1}{2}\mathbf{p}^T M^{-1}\mathbf{p} - \frac{1}{2}\log|M| + \mu(\xi - \bar{\xi})^2/2,\qquad(4)$$

where $\xi$ is the thermostat variable, and $\mu, \bar{\xi}$ are constants chosen to preserve correct marginals. The SGNHT dynamics can used in the E step to maintain the above energy, and we can use the collected **p** samples in the M step as before. We call the resultant method SGNHT-EM, as shown in Alg. 2. Note that, unlike standard HMC above, we do not perform Metropolis-Hastings corrections steps on the gathered samples for these cases. As shown in the algorithms, we collect one set of momenta samples per epoch, after the leapfrog iterations. We use $S\_count$ to denote the number of such samples collected before running an M-step update.

The advantage of this MCEM approach over the parameter-dependent Riemannian variants is twofold:

1. The existing Riemannian adaptive algorithms in the literature [3, 6, 8] all start by modifying the energy function, whereas our framework does not have any such requirement. As long as one uses a sampling mechanism that preserves some energy with correct marginals for $\boldsymbol{\theta}$, in a stochastic sense or otherwise, it can be used in the E step of our framework.

2. The primary disadvantage of the Riemannian algorithms is the added complexity in the dynamics derived from the modified energy functions. One typically ends up using generalized leapfrog dynamics [3, 6], which can lead to implicit systems of equations; to solve these one either has to use standard solvers that have complexity at least cubic in the dimensionality [23, 24], with scalability issues in high dimensional datasets, or use fixed point updates with worsened error guarantees. An alternative approach is to use diagonal covariance matrices, as mentioned earlier, which ignores the coordinate correlations. Our MCEM approach sidesteps all these issues by keeping the existing dynamics of the desired E step sampler unchanged. As shown in the experiments, we can match or beat the Riemannian samplers in accuracy and efficiency by using suitable sample sizes and M step updates, with significantly improved sampling complexities and runtimes.

## 3.2 Dynamic Updates for the E-step Sample Size

---

**Algorithm 1** HMC-EM

---

  **Input:** $\boldsymbol{\theta}^{(0)}, \epsilon, LP\_S, S\_count$
  · Initialize $M$;
  **repeat**
    · Sample $\mathbf{p}^{(t)} \sim N(0, M)$;
    **for** $i = 1$ **to** $LP\_S$ **do**
      · $\mathbf{p}^{(i)} \leftarrow \mathbf{p}^{(i+\epsilon-1)}, \boldsymbol{\theta}^{(i)} \leftarrow \boldsymbol{\theta}^{(i+\epsilon-1)}$;
      · $\mathbf{p}^{(i+\epsilon/2)} \leftarrow \mathbf{p}^{(i)} - \frac{\epsilon}{2}\nabla_{\boldsymbol{\theta}}H(\boldsymbol{\theta}^{(i)}, \mathbf{p}^{(i)})$;
      · $\boldsymbol{\theta}^{(i+\epsilon)} \leftarrow \boldsymbol{\theta}^{(i)} + \frac{\epsilon}{2}\nabla_{\mathbf{p}}H(\boldsymbol{\theta}^{(i)}, \mathbf{p}^{(i+\epsilon/2)})$;
      · $\mathbf{p}^{(i+\epsilon)} \leftarrow \mathbf{p}^{(i+\epsilon/2)} - \frac{\epsilon}{2}\nabla_{\boldsymbol{\theta}}H(\boldsymbol{\theta}^{(i+\epsilon)}, \mathbf{p}^{(i+\epsilon/2)})$;
    **end for**
    · Set $\left(\boldsymbol{\theta}^{(t+1)}, \mathbf{p}^{(t+1)}\right)$ from $\left(\boldsymbol{\theta}^{LP\_S+\epsilon}, \mathbf{p}^{LP\_S+\epsilon}\right)$
      using Metropolis-Hastings
    · Store MC-EM sample $\mathbf{p}^{(t+1)}$;
    **if** $(t+1) \bmod S\_count = 0$ **then**
      · Update $M$ using MC-EM samples;
    **end if**
    · Update $S\_count$ as described in the text;
  **until** forever

---

We now turn our attention to the task of learning the sample size in the E step from the data. The nontriviality of this issue is due to the following reasons: first, we cannot let the sampling dynamics run to convergence in each E step without making the whole process prohibitively slow; second, we have to account for the correlation among successive samples, especially early on in the process when the Markov chain is far from convergence, possibly with "thinning" techniques; and third, we may want to increase the sample count as the chain matures and gets closer to the stationary distribution, and use relatively fewer samples early on.

To this end, we leverage techniques derived from the MCEM literature in statistics [11, 13, 25] to first evaluate a suitable "test" function of the target parameters at certain subsampled steps, using the gathered samples and current M step estimates. We then use confidence intervals created around these evaluations to gauge the relative effect of successive MCEM estimates over the Monte Carlo error. If the updated values of these functions using newer M-step estimates lie in these intervals, we increase the number of samples collected in the next MCEM loop.

Specifically, similar to [13], we start off with the following test function for HMC-EM (Alg. 1): $\mathbf{q}(\cdot) = \left[M^{-1}\mathbf{p}, \nabla\mathcal{L}(\boldsymbol{\theta})\right]$. We then subsample some timesteps as mentioned below, evaluate $\mathbf{q}$ at those steps, and create confidence intervals using sample means and variances:

$$m_S = \frac{1}{S}\sum_{s=1}^{S}\mathbf{q}_s, \quad v_S = \frac{1}{S}\sum_{s=1}^{S}\mathbf{q}_s^2 - m_S^2, \quad C_S := m_S \pm z_{1-\alpha/2}v_S,$$

where $S$ denotes the subsample count, $z_{1-\alpha/2}$ is the $(1-\alpha)$ critical value of a standard Gaussian, and $C_S$ the confidence interval mentioned earlier. For SGNHT-EM (Alg. 2), we use the following test function: $\mathbf{q}(\cdot) = \left[M^{-1}\mathbf{p}, \nabla\mathcal{L}(\boldsymbol{\theta}) + \xi M^{-1}\mathbf{p}, \mathbf{p}^T M^{-1}\mathbf{p}\right]$, derived from the SGNHT dynamics.

---

**Algorithm 2** SGNHT-EM

---

**Input:** $\boldsymbol{\theta}^{(0)}, \epsilon, A, LP\_S, S\_$count
· Initialize $\xi^{(0)}, \mathbf{p}^{(0)}$ and $M$;
**repeat**
   **for** $i = 1$ **to** $LP\_S$ **do**
     · $\mathbf{p}^{(i+1)} \leftarrow \mathbf{p}^{(i)} - \epsilon\xi^{(i)}M^{-1}\mathbf{p}^{(i)} - \epsilon\tilde{\nabla}\mathcal{L}(\boldsymbol{\theta}^{(i)}) +$
       $\sqrt{2A}\mathcal{N}(0,\epsilon)$;
     · $\boldsymbol{\theta}^{(i+1)} \leftarrow \boldsymbol{\theta}^{(i)} + \epsilon M^{-1}\mathbf{p}^{(i+1)}$;
     · $\xi^{(i+1)} \leftarrow \xi^{(i)} + \epsilon\left[\frac{1}{D}\mathbf{p}^{(i+1)T}M^{-1}\mathbf{p}^{(i+1)} - 1\right]$;
   **end for**
   · Set $\left(\boldsymbol{\theta}^{(t+1)}, \mathbf{p}^{(t+1)}, \xi^{(t+1)}\right) =$
    $\left(\boldsymbol{\theta}^{(LP\_S+1)}, \mathbf{p}^{(LP\_S+1)}, \xi^{(LP\_S+1)}\right)$;
   · Store MC-EM sample $\mathbf{p}^{(t+1)}$;
   **if** $(t+1)$ mod $S\_$count $= 0$ **then**
     · Update $M$ using MC-EM samples;
   **end if**
   · Update $S\_$count as described in the text;
**until** forever

---

One can adopt the following method described in [25]: choose the subsampling offsets $\{t_1 \ldots t_S\}$ as $t_s = \sum_{i=1}^{s} x_i$, where $x_i - 1 \sim \text{Poisson}(\nu i^d)$, with suitably chosen $\nu \geq 1$ and $d > 0$. We found both this and a fixed set of $S$ offsets to work well in our experiments.

With the subsamples collected using this mechanism, we calculate the confidence intervals as described earlier. The assumption is that this interval provides an estimate of the spread of $\mathbf{q}$ due to the Monte Carlo error. We then perform the M-step, and evaluate $\mathbf{q}$ using the updated M-step estimates. If this value lies in the previously calculated confidence bound, we increase $S$ as $S = S + S/S_I$ in the following iteration to overcome the Monte Carlo noise. See [11, 13] for details on these procedures. Values for the constants $\nu, \alpha, d, S_I$, as well as initial estimates for $S$ are given in the supplementary. Running values for $S$ are denoted $S\_$count hereafter.

### 3.3 An Online Update for the M-Step

Next we turn our attention to the task of updating the mass matrices using the collected momenta samples. As shown in the energy functions above, the momenta are sampled from zero-mean normal distributions, enabling us to use standard covariance estimation techniques from the literature. However, since we are using discretized MCMC to obtain these samples, we have to address the variance arising from the Monte Carlo error, especially during the burn-in phase. To that end, we found a running average of the updates to work well in our experiments; in particular, we updated the *inverse* mass matrix, denoted as $M_I$, at the $k^{\text{th}}$ M-step as:

$$M_I^{(k)} = (1 - \kappa^{(k)})M_I^{(k-1)} + \kappa^{(k)}M_I^{(k,\text{est})}, \tag{5}$$

where $M_I^{(k,\text{est})}$ is a suitable estimate computed from the gathered samples in the $k^{\text{th}}$ M-step, and $\left\{\kappa^{(k)}\right\}$ is a step sequence satisfying some standard assumptions, as described below. Note that the $M_I$s correspond to the precision matrix of the Gaussian distribution of the momenta; updating this during the M-step also removes the need to invert the mass matrices during the leapfrog iterations. Curiously, we found the inverse of the empirical covariance matrix to work quite well as $M_I^{(k,\text{est})}$ in our experiments.

These updates also induce a fresh perspective on the convergence of the overall MCEM procedure. Existing convergence analyses in the statistics literature fall into three broad categories: a) the almost sure convergence presented in [26] as $t \to \infty$ with increasing sample sizes, b) the asymptotic angle presented in [27], where the sequence of MCEM updates are analyzed as an approximation to the standard EM sequence as the sample size, referred to as $S\_$count above, tends to infinity, and c) the asymptotic consistency results obtained from multiple Gibbs chains in [28], by letting the chain counts and iterations tend to $\infty$. Our analysis differs from all of these, by focusing on the maximum likelihood situations noted above as convex optimization problems, and using SGD convergence techniques [29] for the sequence of iterates $M_I^{(k)}$.

**Proposition 1.** *Assume the $M_I^{(k,est)}$'s provide an unbiased estimate of $\nabla J$, and have bounded eigenvalues. Let $\inf_{\|M_I - M_I^*\|^2 > \epsilon} \nabla J(M_I) > 0 \ \forall \epsilon > 0$. Further, let the sequence $\left\{\kappa^{(k)}\right\}$ satisfy $\sum_k \kappa^{(k)} = \infty$, $\sum_k \left(\kappa^{(k)}\right)^2 < \infty$. Then the sequence $\left\{M_I^{(k)}\right\}$ converges to the MLE of the precision almost surely.*

Recall that the (negative) *precision* is a natural parameter of the normal distribution written in exponential family notation, and that the log-likelihood is a concave function of the natural parameters for this family; this makes max-likelihood a convex optimization problem over the precision, even in the presence of linear constraints [30, 31]. Therefore, this implies that the problems (3), (4) have a unique maximum, denoted by $M_I^*$ above. Also note that the update (5) corresponds to a first order update on the iterates with an L2-regularized objective, with unit regularization parameter; this is denoted by $J(M_I)$ in the proposition. That is, $J$ is the energy preserved by our sampler(s), as a function of the mass (precision), augmented with an L2 regularization term. The resultant strongly convex optimization problem can be analyzed using SGD techniques under the assumptions noted above; we provide a proof in the supplementary for completeness.

We should note here that the "stochasticity" in the proof does not refer to the stochastic gradients of $\mathcal{L}(\boldsymbol{\theta})$ used in the leapfrog dynamics of Algorithms 2 through 5; instead we think of the collected momenta samples as a stochastic minibatch used to compute the gradient of the regularized energy, as a function of the covariance (mass), allowing us to deal with the Monte Carlo error indirectly. Also note that our assumption on the unbiasedness of the $M_I^{(k,\mathrm{est})}$ estimates is similar to [26], and distinct from assuming that the MCEM samples of $\boldsymbol{\theta}$ are unbiased; indeed, it would be difficult to make this latter claim, since stochastic samplers in general are known to have a convergent bias.

### 3.4  Nosé-Poincaré Variants

We next develop a stochastic version of the dynamics derived from the Nosé-Poincaré Hamiltonian, followed by an MCEM variant. This allows for a direct comparison of the Riemann manifold formulation and our MCEM framework for learning the kinetic masses, in a stochastic setting with thermostat controls on the momentum terms and desired properties like reversibility and symplecticness provided by generalized leapfrog discretizations. The Nosé-Poincaré energy function can be written as [6, 9]:

$$H_{NP} = s\left[-\mathcal{L}(\boldsymbol{\theta}) + \frac{1}{2}\left(\frac{\mathbf{p}}{s}\right)M^{-1}\left(\frac{\mathbf{p}}{s}\right) + \frac{q^2}{2Q} + gkT\log s - H_0\right], \tag{6}$$

where $\mathcal{L}(\boldsymbol{\theta})$ is the joint log-likelihood, $s$ is the thermostat control, $\mathbf{p}$ and $q$ the momentum terms corresponding to $\boldsymbol{\theta}$ and $s$ respectively, and $M$ and $Q$ the respective mass terms. See [6, 9] for descriptions of the other constants. Our goal is to learn both $M$ and $Q$ using the MCEM framework, as opposed to [6], where both were formulated in terms of $\boldsymbol{\theta}$. To that end, we propose the following system of equations for the stochastic scenario:

$$\mathbf{p}^{t+\epsilon/2} = \mathbf{p} + \frac{\epsilon}{2}\left[s\tilde{\nabla}\mathcal{L}(\boldsymbol{\theta}) - \frac{B(\boldsymbol{\theta})}{\sqrt{s}}M^{-1}\mathbf{p}^{t+\epsilon/2}\right], \quad \frac{\epsilon}{4Q}(q^{t+\epsilon/2})^2 + \left[1 + \frac{A(\boldsymbol{\theta})s\epsilon}{2Q}\right]q^{t+\epsilon/2}$$

$$- \left[q + \frac{\epsilon}{2}\left[-gkT(1+\log s) + \frac{1}{2}\left(\frac{\mathbf{p}^{t+\epsilon/2}}{s}\right)M^{-1}\left(\frac{\mathbf{p}^{t+\epsilon/2}}{s}\right) + \tilde{\mathcal{L}}(\boldsymbol{\theta}) + H_0\right]\right] = 0,$$

$$s^{t+\epsilon} = s + \epsilon\left[\frac{q^{t+\epsilon/2}}{Q}\left(s + s^{t+\epsilon/2}\right)\right], \quad \boldsymbol{\theta}^{t+\epsilon} = \boldsymbol{\theta} + \epsilon M^{-1}\mathbf{p}\left[\frac{1}{s} + \frac{1}{s^{t+\epsilon}}\right],$$

$$\mathbf{p}^{t+\epsilon} = \mathbf{p}^{t+\epsilon/2} + \frac{\epsilon}{2}\left[s^{t+\epsilon}\tilde{\nabla}\mathcal{L}(\boldsymbol{\theta}^{t+\epsilon}) - \frac{B(\boldsymbol{\theta}^{t+\epsilon})}{\sqrt{s^{t+\epsilon}}}M^{-1}\mathbf{p}^{t+\epsilon/2}\right], \quad q^{t+\epsilon} = q^{t+\epsilon/2} + \frac{\epsilon}{2}\left[H_0 + \tilde{\mathcal{L}}(\boldsymbol{\theta}^{t+\epsilon})\right.$$

$$\left. - gkT(1+\log s^{t+\epsilon}) + \frac{1}{2}\left(\frac{\mathbf{p}^{t+\epsilon/2}}{s^{t+\epsilon}}\right)M^{-1}\left(\frac{\mathbf{p}^{t+\epsilon/2}}{s^{t+\epsilon}}\right) - \frac{A(\boldsymbol{\theta})s^{t+\epsilon}}{2Q}q^{t+\epsilon/2} - \frac{\left(q^{t+\epsilon/2}\right)^2}{2Q}\right], \tag{7}$$

where $t + \epsilon/2$ denotes the half-step dynamics, $\tilde{}$ signifies noisy stochastic estimates, and $A(\boldsymbol{\theta})$ and $B(\boldsymbol{\theta})$ denote the stochastic noise terms, necessary for the Fokker-Planck corrections [6]. Note that we only have to solve a quadratic equation for $q^{t+\epsilon/2}$ with the other updates also being closed-form, as opposed to the implicit system of equations in [6].

**Proposition 2.** *The dynamics* (7) *preserve the Nosé-Poincaré energy* (6).

The proof is a straightforward application of the Fokker-Planck corrections for stochastic noise to the Hamiltonian dynamics derived from (6), and is provided in the supplementary. With these dynamics, we first develop the SG-NPHMC algorithm (Alg. 4 in the supplementary) as a counterpart to SGHMC and SGNHT, and wrap it in our MCEM framework to create SG-NPHMC-EM (Alg. 5 in the supplementary). As we shall demonstrate shortly, this EM variant performs comparably to SGR-NPHMC from [6], while being significantly faster.

# 4 Experiments

In this section we compare the performance of the MCEM-augmented variants of HMC, SGHMC as well as SGNHT with their standard counterparts, where the mass matrices are set to the identity matrix. We call these augmented versions HMC-EM, SGHMC-EM, and SGNHT-EM respectively. As baselines for the synthetic experiments, in addition to the standard samplers mentioned above, we also evaluate RHMC [3] and SGR-NPHMC [6], two recent algorithms based on dynamic Riemann manifold formulations for learning the mass matrices. In the topic modeling experiment, for scalability reasons we evaluate only the stochastic algorithms, including the recently proposed SGR-NPHMC, and omit HMC, HMC-EM and RHMC. Since we restrict the discussions in this paper to samplers with second-order dynamics, we do not compare our methods with SGLD [7] or SGRLD [8].

## 4.1 Parameter Estimation of a 1D Standard Normal Distribution

In this experiment we aim to learn the parameters of a unidimensional standard normal distribution in both batch and stochastic settings, using $5,000$ data points generated from $\mathcal{N}(0,1)$, analyzing the impact of our MC-EM framework on the way. We compare all the algorithms mentioned so far: HMC, HMC-EM, SGHMC, SGHMC-EM, SGNHT, SGNHT-EM, SG-NPHMC, SG-NPHMC-EM along with RHMC and SGR-NPHMC. The generative model consists of normal-Wishart priors on the mean $\mu$ and precision $\tau$, with posterior distribution $p(\mu, \tau | \mathbf{X}) \propto N(\mathbf{X}|\mu, \tau)\mathcal{W}(\tau|1, 1)$, where $\mathcal{W}$ denotes the Wishart distribution. We run all the algorithms for the same number of iterations, discarding the first $5,000$ as "burn-in". Batch sizes were fixed to $100$ for all the stochastic algorithms, along with $10$ leapfrog iterations across the board. For SGR-NPHMC and RHMC, we used the observed Fisher information plus the negative Hessian of the prior as the tensor, with one fixed point iteration on the implicit system of equations arising from the dynamics of both. For HMC we used a fairly high learning rate of $1e-2$. For SGHMC and SGNHT we used $A = 10$ and $A = 1$ respectively. For SGR-NPHMC we used $A, B = 0.01$.

We show the RMSE numbers collected from post-burn-in samples as well as per-iteration runtimes in Table 1. An "iteration" here refers to a complete E step, with the full quota of leapfrog jumps. The improvements afforded by our MCEM framework are immediately noticeable; HMC-EM matches the errors obtained from RHMC, in effect matching the sample distribution, while being much faster (an order of magnitude) per iteration. The stochastic MCEM algorithms show markedly better performance as well; SGNHT-EM in particular beats SGR-NPHMC in RMSE-$\tau$ while being significantly faster due to simpler updates for the mass matrices. Accuracy improvements are particularly noticeable for the high learning rate regimes for HMC, SGHMC and SG-NPHMC.

| METHOD | RMSE $(\mu)$ | RMSE $(\tau)$ | TIME |
|---|---|---|---|
| HMC | 0.0196 | 0.0197 | 0.417MS |
| HMC-EM | 0.0115 | 0.0104 | 0.423MS |
| RHMC | 0.0111 | 0.0089 | 5.748MS |
| SGHMC | 0.1590 | 0.1646 | 0.133MS |
| SGHMC-EM | 0.0713 | 0.2243 | 0.132MS |
| SG-NPHMC | 0.0326 | 0.0433 | 0.514MS |
| SG-NPHMC-EM | 0.0274 | 0.0354 | 0.498MS |
| SGR-NPHMC | 0.0240 | 0.0308 | 3.145MS |
| SGNHT | 0.0344 | 0.0335 | 0.148MS |
| SGNHT-EM | 0.0317 | 0.0289 | 0.148MS |

Table 1: RMSE of the sampled means, precisions and per-iteration runtimes (in milliseconds) from runs on synthetic Gaussian data.

## 4.2 Parameter Estimation in 2D Bayesian Logistic Regression

Next we present some results obtained from a Bayesian logistic regression experiment, using both synthetic and real datasets. For the synthetic case, we used the same methodology as [6]; we generated $2,000$ observations from a mixture of two normal distributions with means at $[1, -1]$ and $[-1, 1]$, with mixing weights set to $(0.5, 0.5)$ and the covariance set to $I$. We then classify these points using a linear classifier with weights $\{W_0, W_1\} = [1, -1]$, and attempt to learn these weights using our samplers. We put $\mathcal{N}(0, 10I)$ priors on the weights, and used the metric tensor described in §7 of [3] for the Riemannian samplers. In the (generalized) leapfrog steps of the Riemannian samplers, we opted to use 2 or 3 fixed point iterations to approximate the solutions to the implicit equations. Along with this synthetic setup, we also fit a Bayesian LR model to the Australian Credit and Heart regression datasets from the UCI database, for additional runtime comparisons. The Australian credit dataset contains $690$ datapoints of dimensionality $14$, and the Heart dataset has $270$ $13$-dimensional datapoints.

For the synthetic case, we discard the first 10,000 samples as burn-in, and calculate RMSE values from the remaining samples. Learning rates were chosen from $\{1e-2, 1e-4, 1e-6\}$, and values of the stochastic noise terms were selected from $\{0.001, 0.01, 0.1, 1, 10\}$. Leapfrog steps were chosen from $\{10, 20, 30\}$. For the stochastic algorithms we used a batchsize of 100.

| METHOD | RMSE ($W_0$) | RMSE ($W_1$) |
|---|---|---|
| HMC | 0.0456 | 0.1290 |
| HMC-EM | 0.0145 | 0.0851 |
| RHMC | 0.0091 | 0.0574 |
| SGHMC | 0.2812 | 0.2717 |
| SGHMC-EM | 0.2804 | 0.2583 |
| SG-NPHMC | 0.4945 | 0.4263 |
| SG-NPHMC-EM | 0.0990 | 0.4229 |
| SGR-NPHMC | 0.1901 | 0.1925 |
| SGNHT | 0.2035 | 0.1921 |
| SGNHT-EM | 0.1983 | 0.1729 |

Table 2: RMSE of the two regression parameters, for the synthetic Bayesian logistic regression experiment. See text for details.

The RMSE numbers for the synthetic dataset are shown in Table 2, and the per-iteration runtimes for all the datasets are shown in Table 3. We used initialized $S\_count$ to 300 for HMC-EM, SGHMC-EM, and SGNHT-EM, and 200 for SG-NPHMC-EM. The MCEM framework noticeably improves the accuracy in almost all cases, with no computational overhead. Note the improvement for SG-NPHMC in terms of RMSE for $W_0$. For the runtime calculations, we set all samplers to 10 leapfrog steps, and fixed $S\_count$ to the values mentioned above.

The comparisons with the Riemannian algorithms tell a clear story: though we do get somewhat better accuracy with these samplers, they are orders of magnitude slower. In our synthetic case, for instance, each iteration of RHMC (consisting of all the leapfrog steps and the M-H ratio calculation) takes more than a second, using 10 leapfrog steps and 2 fixed point iterations for the

| METHOD | TIME (SYNTH) | TIME (AUS) | TIME (HEART) |
|---|---|---|---|
| HMC | 1.435MS | 0.987MS | 0.791MS |
| HMC-EM | 1.428MS | 0.970MS | 0.799MS |
| RHMC | 1550MS | 367MS | 209MS |
| SGHMC | 0.200MS | 0.136MS | 0.112MS |
| SGHMC-EM | 0.203MS | 0.141MS | 0.131MS |
| SG-NPHMC | 0.731MS | 0.512MS | 0.403MS |
| SG-NPHMC-EM | 0.803MS | 0.525MS | 0.426MS |
| SGR-NPHMC | 6.720MS | 4.568MS | 3.676MS |
| SGNHT | 0.302MS | 0.270MS | 0.166MS |
| SGNHT-EM | 0.306MS | 0.251MS | 0.175MS |

Table 3: Per-iteration runtimes (in milliseconds) for Bayesian logistic regression experiments, on both synthetic and real datasets.

implicit leapfrog equations, whereas both HMC and HMC-EM are simpler and much faster. Also note that the M-step calculations for our MCEM framework involve a single-step closed form update for the precision matrix, using the collected samples of **p** once every $S\_count$ sampling steps; thus we can amortize the cost of the M-step over the previous $S\_count$ iterations, leading to negligible changes to the per-sample runtimes.

### 4.3 Topic Modeling using a Nonparametric Gamma Process Construction

Next we turn our attention to a high-dimensional topic modeling experiment using a nonparametric Gamma process construction. We elect to follow the experimental setup described in [6]. Specifically, we use the Poisson factor analysis framework of [32]. Denoting the vocabulary as $V$, and the documents in the corpus as $D$, we model the observed counts of the vocabulary terms as $\mathbf{D}_{V \times N} = \text{Poi}(\mathbf{\Phi\Theta})$, where $\mathbf{\Theta}_{K \times N}$ models the counts of $K$ latent topics in the documents, and $\mathbf{\Phi}_{V \times K}$ denotes the factor load matrix, that encodes the relative importance of the vocabulary terms in the latent topics. Following standard Bayesian convention, we put model the columns of $\mathbf{\Phi}$ as $\phi_{\cdot,k} \sim \text{Dirichlet}(\alpha)$, using normalized Gamma variables: $\phi_{v,k} = \frac{\gamma_v}{\sum_v \gamma_v}$, with $\gamma_v \sim \Gamma(\alpha, 1)$. Then we have $\theta_{n,k} \sim \Gamma(r_k, \frac{p_j}{1-p_j})$; we put $\beta(a_0, b_0)$ priors on the document-specific mixing probabilities $p_j$. We then set the $r_k$s to the atom weights generated by the constructive Gamma process definition of [14]; we refer the reader to that paper for the details of the formulation. It leads to a rich nonparametric construction of this Poisson factor analysis model for which closed-form Gibbs updates are infeasible, thereby providing a testing application area for the stochastic MCMC algorithms. We omit the Metropolis Hastings correction-based HMC and RHMC samplers in this evaluation due to poor scalability.

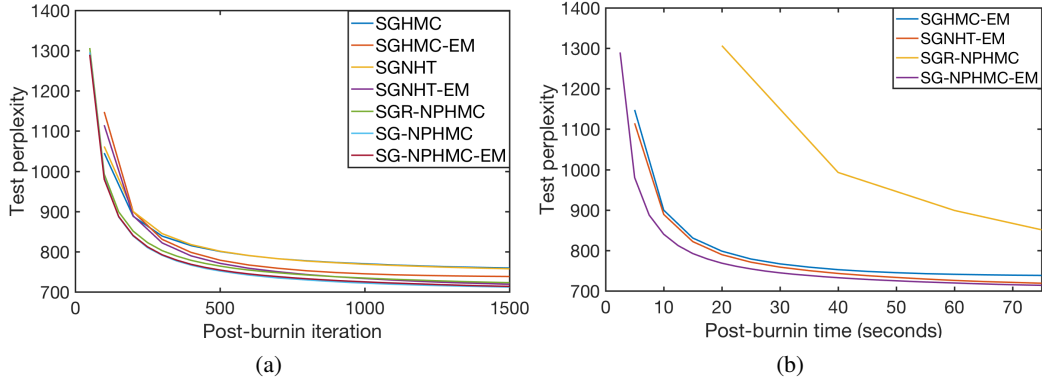

Figure 1: Test perplexities plotted against (a) post-burnin iterations and (b) wall-clock time for the 20-Newsgroups dataset. See text for experimental details.

We use count matrices from the 20-Newsgroups and Reuters Corpus Volume 1 corpora [33]. The former has $2,000$ words and $18,845$ documents, while the second has a vocabulary of size $10,000$ over $804,414$ documents. We used a chronological $60-40$ train-test split for both datasets. Following standard convention for stochastic algorithms, following each minibatch we learn document-specific parameters from $80\%$ of the test set, and calculate test perplexities on the remaining $20\%$. Test perplexity, a commonly used measure for such evaluations, is detailed in the supplementary.

As noted in [14], the atom weights have three sets of components: the $E_k$s, $T_k$s and the hyperparameters $\alpha, \gamma$ and $c$. As in [6], we ran three parallel chains for these parameters, collecting samples of the momenta from the $T_k$ and hyperparameter chains for the MCEM mass updates. We kept the mass of the $E_k$ chain fixed to $I_K$, and chose $K = 100$ as number of latent topics. We initialized S_count, the E-step sample size in our algorithms, to 50 for NPHMC-EM and 100 for the rest. Increasing S_count over time yielded fairly minor improvements, hence we kept it fixed to the values above for simplicity. Additional details on batch sizes, learning rates, stochastic noise estimates, leapfrog iterations etc are provided in the supplementary. For the 20-Newsgroups dataset we ran all algorithms for $1,500$ burn-in iterations, and collected samples for the next $1,500$ steps thereafter, with a stride of 100, for perplexity calculations. For the Reuters dataset we used $2,500$ burn-in iterations. Note that for all these algorithms, an "iteration" corresponds to a full E-step with a stochastic minibatch.

The numbers obtained at the end of the runs are shown in Table 2, along with per-iteration runtimes. The post-burnin perplexity-vs-iteration plots from the 20-Newsgroups dataset are shown in Figure 1. We can see significant improvements from the MCEM framework for all samplers, with that of SGNHT being highly pronounced (719 vs 757); indeed, the SG-NPHMC samplers have

| METHOD | 20-NEWS | REUTERS | TIME(20-NEWS) |
|---|---|---|---|
| SGHMC | 759 | 996 | 0.047s |
| SGHMC-EM | 738 | 972 | 0.047s |
| SGNHT | 757 | 979 | 0.045s |
| SGNHT-EM | 719 | 968 | 0.045s |
| SGR-NPHMC | 723 | 952 | 0.410s |
| SG-NPHMC | 714 | 958 | 0.049s |
| SG-NPHMC-EM | 712 | 947 | 0.049s |

Table 4: Test perplexities and per-iteration runtimes on 20-Newsgroups and Reuters datasets.

lower perplexities (712) than those obtained by SGR-NPHMC (723), while being close to an order of magnitude faster per iteration for 20-Newsgroups even when the latter used diagonalized metric tensors, ostensibly by avoiding implicit systems of equations in the leapfrog steps to learn the kinetic masses. The framework yields nontrivial improvements for the Reuters dataset as well.

## 5  Conclusion

We propose a new theoretically grounded approach to learning the mass matrices in Hamiltonian-based samplers, including both standard HMC and stochastic variants, using a Monte Carlo EM framework. In addition to a newly proposed stochastic sampler, we augment certain existing samplers with this technique to devise a set of new algorithms that learn the kinetic masses dynamically from the data in a flexible and scalable fashion. Experiments conducted on synthetic and real datasets demonstrate the efficacy and efficiency of our framework, when compared to existing Riemannian manifold-based samplers.

**Acknowledgments**

We thank the anonymous reviewers for their insightful comments and suggestions. This material is based upon work supported by the National Science Foundation under Grant No. DMS-1418265. Any opinions, findings, and conclusions or recommendations expressed in this material are those of the author(s) and do not necessarily reflect the views of the National Science Foundation.

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
