[Supplementary Material · hmcem_supp_nips2017.pdf]

# Adaptive Bayesian Sampling with Monte Carlo EM

**Anirban Roychowdhury,   Srinivasan Parthasarathy**
Department of Computer Science and Engineering
The Ohio State University
roychowdhury.7@osu.edu, srini@cse.ohio-state.edu

## 1   Appendices

### 1.1   Proposition 1: Convergence Discussion

We propose the following update for the precision / inverse mass matrix, denoted as $M_I$, at the $k^{\text{th}}$ M-step:

$$M_I^{(k)} = (1 - \kappa^{(k)})M_I^{(k-1)} + \kappa^{(k)}M_I^{(k,\text{est})}, \tag{1}$$

where $M_I^{(k,\text{est})}$ is the estimate computed from the gathered samples in the $k^{\text{th}}$ M-step, and $\left\{\kappa^{(k)}\right\}$ is a step sequence satisfying some standard assumptions, as described below.

**Proposition 1.** *Assume the $M_I^{(k,est)}$'s provide an unbiased estimate of $\nabla J$, and have bounded eigenvalues. Let $\inf_{\|M_I - M_I^*\|^2 > \epsilon} \nabla J(M_I) > 0 \ \forall \epsilon > 0$. Further, let the sequence $\left\{\kappa^{(k)}\right\}$ satisfy $\sum_k \kappa^{(k)} = \infty, \sum_k \left(\kappa^{(k)}\right)^2 < \infty$. Then the sequence $\left\{M_I^{(k)}\right\}$ converges to the MLE of the precision almost surely.*

*Proof.* The proof follows the basic outline laid out in [1]. With a slight abuse of notation, we use $M_k$ to denote the iterates, $\bar{M}_k$ to denote $M_I^{(k,\text{est})}$, and replace the $\kappa^{(k)}$s with $\kappa_k$. Then, as mentioned in the main text, the update (1) can be written in the following first-order form:

$$M_k = M_{k-1} + \kappa_k \nabla J(M_k),$$

where $J(\cdot)$ is the L2-regularized energy mentioned in the main text, as a function of the precision, and we assume $\mathbb{E}_z \bar{M}_k(z) = \nabla J(M_k)$, $z$ being a random variable codifying the stochasticity in the estimate $\bar{M}_k$. As mentioned in the main paper, this stochasticity can be thought of as a surrogate for the Monte Carlo error in the collected momenta samples. Now define the Lyapunov function:

$$h(M_k) = \|M_k - M^*\|^2,$$

where $M^*$ is the unique maximizer of the regularized objective function; as mentioned earlier, this exists because the precision is a natural parameter of the normal written in exponential family form, and the log likelihood of the latter is concave in the natural parameters. Then we can write the difference in Lyapunov errors for successive iterates as

$$h(M_{k+1}) - h(M_k) = -2\kappa_k \left(M_k - M^*\right)^T \bar{M}_k(z_k) + \kappa_k^2 \|\bar{M}_k(z_k)\|^2.$$

Denoting the $\sigma$-algebra of all the $z$ variables seen till the $k^{\text{th}}$ step by $\mathcal{F}_k$, and using conditional independences of the expectations given this information, we can write the expectation of the quantity above as:

$$\mathbb{E}\left(h(M_{k+1}) - h(M_k)\middle|\,\mathcal{F}_k\right) = -2\kappa_k \left(M_k - M^*\right)^T \nabla J(M_k) + \kappa_k^2 \mathbb{E}\|\bar{M}_k(z_k)\|^2. \tag{2}$$

Now, since we assumed the $\bar{M}_k$'s to have bounded eigenvalues, we can bound the expectation on the right above as follows:

$$\mathbb{E}\|\bar{M}_k(z_k)\|^2 \le A + B\|M_k - M^*\|^2,$$

for sufficiently large values of $A, B \geq 0$. This allows to write 2 as follows:

$$\mathbb{E}\left(h(M_{k+1}) - (1 - \kappa_k^2 B)h(M_k)|\mathcal{F}_k\right) \leq -2\kappa_k\left(M_k - M^*\right)^T \nabla J(M_k) + \kappa_k^2 A. \tag{3}$$

Now we define two sequences as follows:

$$\mu_k = \prod_{i=1}^{k} \frac{1}{1 - \kappa_k^2 B}, \quad h_k' = \mu_k h(M_k). \tag{4}$$

The sequence $\{\mu_k\}$ can be seen to converge based on our assumptions on $\kappa_k^2$. Then we can bound the positive variations of $h_k'$-s as:

$$\mathbb{E}\left[\mathbb{E}(h_{k+1}' - h_k')^+\right]|\mathcal{F}_k \leq \kappa_k^2 \mu_k A.$$

This proves $h_k'$ to be a quasi-martingale. By the convergence theorem for quasi-martingales [2], we know that these converge almost surely. Since $\{\mu_k\}$ converge as well, we have almost sure convergence of the $h(M_k)$'s. Combined with the assumption that $\sum_k \kappa_k = \infty$ and eqn. (3), we have almost sure convergence of $(M_k - M^*)^T \nabla J(M_k)$ to 0. The final assumption of the proposition allows us to use this result to prove that $M_k \to M^*$ almost surely.

$\square$

## 1.2 Stochastic samplers with MCEM augmentations

In this section we present the MCEM variant of the SGHMC algorithm [3], followed by the SG-NPHMC algorithm using stochastic dynamics derived from the Nosé-Poincaré Hamiltonian. This is then given the MCEM treatment, leading to the SG-NPHMC-EM method.

### 1.2.1 SGHMC-EM

The MCEM variant of the SGHMC algorithm, which we denote SGHMC-EM, is given in Alg. (3). We simply take the standard HMC dynamics, add Fokker-Planck correction terms to handle the stochastic noise, and use the MCEM framework from the main paper to collect appropriate number of samples of $\mathbf{p}$, and use them to update the mass $M$. $C$ and $\hat{B}$ are user-specified estimates of the noise in the stochastic gradients.

---

**Algorithm 3** SGHMC-EM

---

   **Input:** $\boldsymbol{\theta}^{(0)}, \epsilon, A, LP\_S, S\_\text{count}$
   · Initialize $\xi^{(0)}, \mathbf{p}^{(0)}$ and $M$;
   **repeat**
      · Sample $\mathbf{p}^{(t)} \sim N(0, M)$;
      **for** $i = 1$ **to** $LP\_S$ **do**
         · $\mathbf{p}^{(i+1)} \leftarrow \mathbf{p}^{(i)} - \epsilon C M^{-1}\mathbf{p}^{(i)} - \epsilon\tilde{\nabla}\mathcal{L}(\boldsymbol{\theta}^{(i)}) + \sqrt{2(C - \hat{B})}\mathcal{N}(0, \epsilon)$;
         · $\boldsymbol{\theta}^{(i+1)} \leftarrow \boldsymbol{\theta}^{(i)} + \epsilon M^{-1}\mathbf{p}^{(i+1)}$;
      **end for**
      · Set $\left(\boldsymbol{\theta}^{(t+1)}, \mathbf{p}^{(t+1)}\right) = \left(\boldsymbol{\theta}^{(LP\_S+1)}, \mathbf{p}^{(LP\_S+1)}\right)$;
      · Store MC-EM sample $\mathbf{p}^{(t+1)}$;
      **if** $(t + 1) \mod S\_\text{count} = 0$ **then**
         · Update $M$ using MC-EM samples;
      **end if**
      · Update $S\_\text{count}$ as described in the text;
   **until** forever

---

### 1.2.2 SG-NPHMC

As mentioned in the main paper, the Nosé-Poincaré energy function can be written as follows [4, 5]:

$$H_{NP} = s\left[-\mathcal{L}(\boldsymbol{\theta}) + \frac{1}{2}\left(\frac{\mathbf{p}}{s}\right)M^{-1}\left(\frac{\mathbf{p}}{s}\right) + \frac{q^2}{2Q} + gkT\log s - H_0\right], \tag{5}$$

where $\mathcal{L}(\boldsymbol{\theta})$ is the joint log-likelihood, $s$ is the thermostat control, $\mathbf{p}$ and $q$ the momentum terms corresponding to $\boldsymbol{\theta}$ and $s$ respectively, and $M$ and $Q$ the respective mass terms. See [4, 5] for descriptions of the other constants. Our goal is to learn both $M$ and $Q$ using the MCEM framework, as opposed to [4], where both were formulated in terms of $\boldsymbol{\theta}$. To that end, we propose the following system of equations for the stochastic scenario:

$$\mathbf{p}^{t+\epsilon/2} = \mathbf{p} + \frac{\epsilon}{2}\left[s\tilde{\nabla}\mathcal{L}(\boldsymbol{\theta}) - \frac{B(\boldsymbol{\theta})}{\sqrt{s}}M^{-1}\mathbf{p}^{t+\epsilon/2}\right], \quad \frac{\epsilon}{4Q}(q^{t+\epsilon/2})^2 + \left[1 + \frac{A(\boldsymbol{\theta})s\epsilon}{2Q}\right]q^{t+\epsilon/2} - \left[q + \frac{\epsilon}{2}\right[$$

$$- gkT(1 + \log s) + \frac{1}{2}\left(\frac{\mathbf{p}^{t+\epsilon/2}}{s}\right)M^{-1}\left(\frac{\mathbf{p}^{t+\epsilon/2}}{s}\right) + \tilde{\mathcal{L}}(\boldsymbol{\theta}) + H_0\right]\right] = 0,$$

$$s^{t+\epsilon} = s + \epsilon\left[\frac{q^{t+\epsilon/2}}{Q}\left(s + s^{t+\epsilon/2}\right)\right], \quad \boldsymbol{\theta}^{t+\epsilon} = \boldsymbol{\theta} + \epsilon M^{-1}\mathbf{p}\left[\frac{1}{s} + \frac{1}{s^{t+\epsilon}}\right],$$

$$\mathbf{p}^{t+\epsilon} = \mathbf{p}^{t+\epsilon/2} + \frac{\epsilon}{2}\left[s^{t+\epsilon}\tilde{\nabla}\mathcal{L}(\boldsymbol{\theta}^{t+\epsilon}) - \frac{B(\boldsymbol{\theta}^{t+\epsilon})}{\sqrt{s^{t+\epsilon}}}M^{-1}\mathbf{p}^{t+\epsilon/2}\right], \quad q^{t+\epsilon} = q^{t+\epsilon/2} + \frac{\epsilon}{2}\left[H_0 + \tilde{\mathcal{L}}(\boldsymbol{\theta}^{t+\epsilon})\right.$$

$$\left. - gkT(1 + \log s^{t+\epsilon}) + \frac{1}{2}\left(\frac{\mathbf{p}^{t+\epsilon/2}}{s^{t+\epsilon}}\right)M^{-1}\left(\frac{\mathbf{p}^{t+\epsilon/2}}{s^{t+\epsilon}}\right) - \frac{A(\boldsymbol{\theta})s}{2Q}q^{t+\epsilon/2} - \frac{(q^{t+\epsilon/2})^2}{2Q}\right],$$

$$\tag{6}$$

where $t + \epsilon/2$ denotes the half-step dynamics, $\tilde{\phantom{x}}$ signifies noisy stochastic estimates, and $A(\boldsymbol{\theta})$ and $B(\boldsymbol{\theta})$ denote the stochastic noise terms, necessary for the Fokker-Planck corrections [4].

**Proposition 2.** *The dynamics* (6) *preserve the Nosé-Poincaré energy* (5).

*Proof.* We start off with the basic dynamics derived from the Nosé-Poincaré Hamiltonian:

$$\dot{\boldsymbol{\theta}} = M^{-1}\frac{\mathbf{p}}{s}$$
$$\dot{\mathbf{p}} = s\nabla\mathcal{L}(\boldsymbol{\theta})$$
$$\dot{s} = \frac{q}{Q}s \tag{7}$$
$$\dot{q} = \mathcal{L}(\boldsymbol{\theta}) + \frac{1}{2}\left(\frac{\mathbf{p}}{s}\right)^T M^{-1}\left(\frac{\mathbf{p}}{s}\right) - gkT(1 + \log s) - \frac{q^2}{2Q} + H_0,$$

where the dot notation denotes the time derivatives. Following the notation of [6], this can be expressed as:

$$\begin{bmatrix} \dot{\boldsymbol{\theta}} \\ \dot{\mathbf{p}} \\ \dot{s} \\ \dot{q} \end{bmatrix} = -\begin{bmatrix} 0 & 0 & 0 & -I \\ 0 & 0 & I & 0 \\ 0 & -I & 0 & 0 \\ I & 0 & 0 & 0 \end{bmatrix}\begin{bmatrix} \frac{\partial}{\partial s}H_{NP} \\ \frac{\partial}{\partial q}H_{NP} \\ \frac{\partial}{\partial \boldsymbol{\theta}}H_{NP} \\ \frac{\partial}{\partial \mathbf{p}}H_{NP} \end{bmatrix} + \mathbf{N}, \tag{8}$$

where $\mathbf{N} = [0, \mathcal{N}(0, 2\sqrt{s}B(\boldsymbol{\theta})), 0, \mathcal{N}(0, 2B(\boldsymbol{\theta}))]$ would be the stochastic noise from the minibatch estimates of $\nabla\mathcal{L}(\boldsymbol{\theta})$ and $\mathcal{L}(\boldsymbol{\theta})$ respectively. Denoting the first matrix on the right by $D$ and the second by $\nabla H_{NP}$, we can see that $\text{tr}\left\{\nabla^T\nabla Dy\right\} = 0$ for any $y = y(\boldsymbol{\theta}, \mathbf{p}, s, q)$.

Recall that the joint distribution of interest, $p(\boldsymbol{\theta}, \mathbf{p}, s, q) \propto \exp(-H_{NP})$; thus $\nabla p(\boldsymbol{\theta}, \mathbf{p}, s, q) = -p\nabla H_{NP}$.

Now, for any stochastic differential equation written as $\dot{\boldsymbol{\theta}} = f(\boldsymbol{\theta}) + \mathcal{N}(0, 2Q(\boldsymbol{\theta}))$, the Fokker-Planck equation can be written as:

$$\frac{\partial}{\partial t}p(\boldsymbol{\theta}) = -\frac{\partial}{\partial \boldsymbol{\theta}}[f(\boldsymbol{\theta})p(\boldsymbol{\theta})] + \frac{\partial^2}{\partial \boldsymbol{\theta}^2}[Q(\boldsymbol{\theta})p(\boldsymbol{\theta})],$$

where $p(\boldsymbol{\theta})$ denotes the distribution of $\boldsymbol{\theta}$, and $\frac{\partial^2}{\partial \boldsymbol{\theta}^2} = \sum_{i,j}\frac{\partial}{\partial \theta_i}\frac{\partial}{\partial \theta_j}$. For our Nosé-Poincaré case, the right hand side can be written as:

$$\text{tr}\left\{\nabla^T X\nabla p(\boldsymbol{\theta}, \mathbf{p}, s, q)\right\} + \text{tr}\nabla^T\left\{p(\boldsymbol{\theta}, \mathbf{p}, s, q)D\nabla H_{NP}\right\}$$
$$= \text{tr}\left\{(X + D)\nabla^T\nabla p(\boldsymbol{\theta}, \mathbf{p}, s, q)\right\} + \text{tr}\nabla^T\left\{p(\boldsymbol{\theta}, \mathbf{p}, s, q)D\nabla H_{NP}\right\},$$

where the diffusion noise matrix

$$X = \begin{bmatrix} 0 & 0 & 0 & 0 \\ 0 & 0 & 0 & \sqrt{s}B(\boldsymbol{\theta}) \\ 0 & 0 & 0 & 0 \\ 0 & A(\boldsymbol{\theta}) & 0 & 0 \end{bmatrix}.$$

Thus replacing $D$ by $X + D$ in (8) would make the RHS zero. This transformation would add correction terms to the dynamics 7 to yield the following:

$$\dot{\boldsymbol{\theta}} = M^{-1}\frac{\mathbf{p}}{s}$$

$$\dot{\mathbf{p}} = s\nabla\mathcal{L}(\boldsymbol{\theta}) - \sqrt{s}B(\boldsymbol{\theta})M^{-1}\frac{\mathbf{p}}{s}$$

$$\dot{s} = \frac{q}{Q}s$$

$$\dot{q} = \mathcal{L}(\boldsymbol{\theta}) + \frac{1}{2}\left(\frac{\mathbf{p}}{s}\right)^T M^{-1}\left(\frac{\mathbf{p}}{s}\right) - gkT(1 + \log s) - \frac{q^2}{2Q} - A(\boldsymbol{\theta})\frac{q}{Q}s + H_0.$$

Discretizing this system using the generalized leapfrog technique gives rise to the dynamics 6.

$\square$

The dynamics 6 therefore induce the SG-NPHMC algorithm, shown in Alg. (4).

---

**Algorithm 4** SG-NPHMC

---

**Input:** $\boldsymbol{\theta}^{(0)}, \epsilon, A, LP\_S, S\_\text{count}$
· Initialize $\mathbf{p}^{(0)}, M, Q$;
**repeat**
  · Sample $\mathbf{p}^{(t)} \sim N(0, M), q \sim N(0, Q)$;
  **for** $i = 1$ **to** $LP\_S$ **do**
    · Perform generalized leapfrog dynamics (6) to get $\mathbf{p}^{(i+\epsilon)}, \boldsymbol{\theta}^{(i+\epsilon)}, s^{(i+\epsilon)}, q^{(i+\epsilon)}$;
  **end for**
  · Set $\left(\boldsymbol{\theta}^{(t+1)}, \mathbf{p}^{(t+1)}, \xi^{(t+1)}\right) = \left(\boldsymbol{\theta}^{(LP\_S+\epsilon)}, \mathbf{p}^{(LP\_S+\epsilon)}, s^{(LP\_S+\epsilon)}, q^{(LP\_S+\epsilon)}\right)$;
**until** forever

---

### 1.2.3 SG-NPHMC-EM

In this section we add the MCEM framework to Alg. (4) above. This allows us to learn $M$ adaptively while preserving the thermostat controls and symplecticness of the SG-NPHMC sampler.

---

**Algorithm 5** SG-NPHMC-EM

---

**Input:** $\boldsymbol{\theta}^{(0)}, \epsilon, A, B, LP\_S, S\_\text{count}$
· Initialize $s^{(0)}, \mathbf{p}^{(0)}, q^{(0)}, M$ and $Q$;
**repeat**
  **for** $i = 1$ **to** $LP\_S$ **do**
    · Perform generalized leapfrog dynamics (6) to get $\mathbf{p}^{(i+\epsilon)}, \boldsymbol{\theta}^{(i+\epsilon)}, s^{(i+\epsilon)}, q^{(i+\epsilon)}$;
  **end for**
  · Set $\left(\boldsymbol{\theta}^{(t+1)}, \mathbf{p}^{(t+1)}, s^{(t+1)}, q^{(t+1)}\right) = \left(\boldsymbol{\theta}^{(LP\_S+\epsilon)}, \mathbf{p}^{(LP\_S+\epsilon)}, s^{(LP\_S+\epsilon)}, q^{(LP\_S+\epsilon)}\right)$;
  · Store MC-EM samples $\mathbf{p}^{(t+1)}$ and $q^{(t+1)}$;
  **if** $(t+1) \bmod S\_\text{count} = 0$ **then**
    · Update $M, Q$ using MC-EM samples of $\mathbf{p}$ and $q$ respectively;
  **end if**
  · Update $S\_\text{count}$ as described in the text;
**until** forever

---

## 1.3 Experimental addenda

For the topic modeling case, we used the following perplexity measure, as defined as [7]:

$$\text{Perplexity} = \exp\left(-\frac{1}{Y}\sum_{n=1}^{N_{\text{test}}}\sum_{v=1}^{V} y_{nv}\log m_{nv}\right),$$

where $y_{nv}$ refers to the count of vocabulary item $v$ in held-out test document $n$, $Y = \sum_{n=1}^{N_{\text{test}}}\sum_{v=1}^{V} y_{nv}$, and $m_{nv} = \sum_{s=1}^{S}\sum_{k=1}^{K}\phi_{vk}^{(s)}\theta_{kn}^{(s)} / \sum_{v=1}^{V}\sum_{s=1}^{S}\sum_{k=1}^{K}\phi_{vk}^{(s)}\theta_{kn}^{(s)}$, where we collect $S$ samples of $\theta, \phi$, and have $K$ latent topics. For the 20-Newsgroups dataset, we used learning tates of $1e-7$ for the $T_k$ chain, $1e-6$ for the hyperparameter chain, for all the samplers. Stochastic noise estimates were of the order of $1e-2$ for SGHMC, SGNHT and their EM variants, and of the order of $1e-1$ for SG-NPHMC and its EM version. We used minibatches of size 100, and 10 leapfrog iterations for all algorithms. The document-level $\theta, \phi$ were learnt using 20 leapfrog iterations of RHMC [8], which we found to mix slightly better than Gibbs.

For the sample size updates, we used $\nu = 1$, $\alpha = 1$, $d = 2$, $S_I = 10$. We initialized S_count to 50 for the topic modeling experiments with SG-NPHMC-EM, 100 for all other cases. All experiments were run on a Macbook pro with a 2.5Ghz core i7 processor and 16GB ram.