[Reviews · NeurIPS 2017]

Reviewer 1



This paper address the task of learning the mass matrices in samplers obtained from dynamics that preserve some energy function, especially the Hamiltonian Monte Carlo method. While previous works make use of the Riemannian preconditioning techniques, the paper suggests an alternative of using Monte Carlo EM to learn the mass matrix from a maximum likelihood problem. The experiments show that proposed sampler achieve comparable results to existing works, while being significantly faster. I suggest the authors provide more intuitions and backgrounds about the connection between maximization of the likelihood of given samples, i.e., equation (3), and the performance of Monte Carlo method. If one accepts this, other parts in Section 3.2 and Section 3.3 make sense for me. In the experiments, comparison of time per iteration seems unnatural since only the E-step of the proposed algorithm is considered. Especially, one can see that SGHMC, SG-NPHMC are slightly slower than SGHMC-EM and SG-NPHMC-EM, which seems strange to me. Moreover, more comparison between update of mass matrix using Monte Carlo EM and Riemannian manifold is necessary, e.g., HMC and RGMC. The scale of the first experiment to small and second experiment is not persuasive enough since the only algorithm using Riemannian manifold, SGR-NPHMC, performs even worse than the one without the update of mass matrix, i.e. SG-NPHMC. The authors should give a mid-scale experiment where they can compare HMC-EM and RHMC directly, in order to compare the effect of Monte Carlo EM update of mass matrix. Minor: Figure 1-b), post-burnin time should be changed to wall-clock time.

Reviewer 2



The paper embeds an HMC method within an MCEM framework. The iterative nature of the MCEM algorithm allows for online adaptation of the mass matrix $M$ along the EM step, by using the momentum samples derived in the E-step to update the mass matrix at the M-step; this happens via a Robbins-Monro-type of adaptation in equation (5) of the paper. The authors also show a method for adapting the MC-sample size using familiar techniques from the MCEM literature. The crux of the article is found in pages 8-11 where all the above are described in details. Applications and comparisons with competing methods are shown, involving a trivial Gaussian target and an interesting non-parametric application. \section*{Comments} The paper is well-written and easy to follow. I find the method interesting. There is no great novelty, but I think that the proposed algorithm can be practically useful. \begin{itemize} \item Can the authors please explain a bit better the connection of their approach with convex optimisation over the mass matrix? (How is the ``MLE of the precision'' defined?) \end{itemize}

Reviewer 3



This paper considers the problem of learning the mass matrix within Hamiltonian type MCMC samplers. The authors do so by proposing the use of an online EM style algorithm with a Monte Carlo sampling step that adjusts according to the sampling error estimates. The issue of how to sample efficiently from complex target densities is a very important one and the authors make a very nice contribution in this paper. It is rather densely written but the topics are well explained with thorough referencing to the existing literature. The idea of making use of EM to estimate the mass matrix in these types of algorithms is very interesting, the proposed algorithms are explained well and appear to perform well in practice. I therefore only have the following relatively minor comments: Line 128: “we perform generalised leapfrog steps” - it seems to me that standard leapfrog integrator is used here, rather than the generalised one? Algorithm 1: I find the use of \epsilon in the algorithm description slightly confusing. Perhaps you could use an alternative notation here to make clear that these integration steps are only the proposed points for the subsequent Metropolis-Hastings step? Line 328: typo “we put model the columns” Section 4.1: Why was the observed FI used here instead of the expected FI? I think the expected FI might give better performance. More generally, this example seems to be a bit of a straw man, given that there isn’t any strong correlation here in the posterior density - a simplified Riemannian MALA sampler would likely outperform any of these HMC methods, which makes the comparison rather moot. More interesting would be a banana shaped distribution, which is where Riemannian MCMC methods are really a bit more useful. More generally, it is not clear to me how well the EM versions would work with strong correlations that change over the posterior. Since it will be converging to a single mass matrix, then I suspect they won’t work well in strongly banana-like cases. This isn’t explored in the paper at all, which is a shame - I think an example like this would be much more illustrative and informative than the current (trivial) Gaussian example.